# The Role of Community Health Workers in the Health and Well-Being of Vulnerable Older Adults during the COVID Pandemic

**DOI:** 10.3390/ijerph20042766

**Published:** 2023-02-04

**Authors:** Matthew Hodges, Dawn Butler, Ariel Spaulding, Debra K. Litzelman

**Affiliations:** 1Department of Medicine, Indiana University School of Medicine, Indianapolis, IN 46202, USA; 2Regenstrief Institute, Inc., Indianapolis, IN 46202, USA; 3Indiana University Health Physicians, Indianapolis, IN 46204, USA

**Keywords:** older adults, wellness, health promotion, community health worker, social determinants of health, social needs, emotional support

## Abstract

The COVID-19 pandemic disrupted social support networks as well as resource access for participants. The purpose of this study was to: analyze the experiences of older adults enrolled in a geriatric-focused community health worker (CHW) support program, to gain a better understanding of how CHWs might enhance care delivery, and to further understand how COVID-19 affected the social and emotional needs and well-being of older adults during the first 18 months of the pandemic. Qualitative analysis was performed on notes entered by CHWs based on 793 telephone encounters with 358 participants between March 2020 and August 2021. Analysis was performed by two reviewers independently coding the data. Weighing the benefits of seeing family against the risks of COVID exposure was a source of emotional distress for participants. Our qualitative analysis suggests that CHWs were effective in providing emotional support and connecting participants to resources. CHWs are capable of bolstering the support networks of older adults and carrying out some of the responsibilities conventionally fulfilled by family supports. CHWs addressed participant needs that are frequently unmet by healthcare team members and provided emotional support to participants contributing to health and well-being. CHW assistance can fill gaps in support left by the healthcare system and family support structures.

## 1. Introduction

Adults over the age of 65 years face a unique set of biopsychosocial challenges. Compared to younger adults, older adults generally experience diminished levels of personal and community mobility [1,2]. Diminished mobility can make it difficult for individuals to remain independent and access resources. Limited personal mobility can make it difficult to complete daily household tasks, while limited community mobility can make it difficult to access facilities such as grocery stores, pharmacies, and medical centers. Older adults also experience higher rates of social isolation compared to the general population. Social isolation and/or a lack of social support in older adults is associated with a myriad of negative mental and physical health outcomes including cognitive decline, depression, hypertension, obesity [3], and overall diminished perceived wellness and engagement in health-promoting behaviors [4,5]. In addition to these direct adverse effects, social isolation can also exacerbate issues related to resource access caused by diminished mobility [6].

The COVID-19 pandemic posed a direct threat to the health of older adults, with the vast majority of COVID deaths occurring among individuals aged 65 and older [7]. In addition to the morbidity and mortality directly caused by the disease, precautions put into place during the early days of the pandemic (e.g., social distancing) further complicated wellness and engagement in the health-promoting behaviors of older adults. Social distancing guidelines, by definition, limited community mobility and increased levels of social isolation among the general population. Since older adults were already at a higher risk for social isolation and diminished mobility than the general population, the effect of early-stage COVID-19 precautions on the mental health, social health, access to healthy diets, opportunities for exercise, and resource access patterns of older adults were of particular concern [8].

The Indiana Geriatrics Education and Training Center (GETC) is an HRSA-funded program designed to improve well-being and health outcomes in older adults by building healthcare systems that equip healthcare providers to address the complex needs of older adults and promote patient and family engagement in care [9]. The Indiana GETC takes an integrated approach to caring for older adults, streamlining services between primary care providers, geriatric specialists, and community-based partner organizations. This model includes educational programs for patients, their families, and providers. Indiana GETC also offers Geriatrics Care (G-Care), a community health worker (CHW)-based resource navigation program for older adults. Roughly half of G-Care participants were from racial/ethnic minority groups, and over 70% of G-care participants were women. Additionally, G-Care participants were recruited from local federally qualified health centers (FQHC) that conventionally serve low-income patients. FQHC are designated by the United States government as outpatient clinics in medically underserved areas and/or by providing health care for underserved populations, thereby qualifying for specific reimbursement under Medicare and Medicaid.

Multiple systematic reviews have found that the integration of CHWs into the patient care team can be an effective way to improve outcomes and reduce health disparities within the context of the US healthcare system [10,11,12]. CHW responsibilities can vary widely between programs, but CHW responsibilities frequently include providing patient education, connecting the patient to local resources, assisting the patient in navigating the healthcare system, and providing emotional support. The potential role of CHWs in providing care to older adults remains relatively understudied compared to the general population. A 2021 systematic review by Kennedy et al. found that CHW-based interventions may be efficacious in addressing the complex health needs of older adults, but additional evidence is necessary [13]. An editorial published in the Journal of the American Geriatrics Society in response to the Kennedy review encouraged the dissemination of data from other HRSA-funded Geriatric Workforce Enhancement Programs as a critical next step in evaluating the role of CHWs in geriatric care [14].

Due to the complex needs of older adults and the variance in populations that they serve, there is a large degree of heterogeneity among CHW-based geriatric programs in terms of the types of services offered and methods of delivery. G-Care CHWs were trained to provide health education to patients who enrolled (participants), connect participants to local resources, assist participants with the logistics of complex procedures (e.g., moving), and provide social support. While initially designed as an in-person home visitation service, COVID-era restrictions led to the program shifting to a mainly telephone-based model. The purpose of this report is to analyze the experiences of older adults enrolled in a CHW support program in order to gain a better understanding of the ways in which CHWs might be able to enhance participants’ well-being and care delivery. Additionally, we sought to further understand the impact of COVID-19 on the lived experiences of older adults during the first 18 months of the pandemic. 

## 2. Materials and Methods

### 2.1. G-Care CHW Training

Two G-Care CHWs were hired and trained by both the clinic staff and the Indiana GETC team. Training (summarized in Table 1 below) included education on common geriatric syndromes (falls, depression, cognitive impairment, risky medication management), the 4Ms of the age-friendly framework (Mentation, Mobility, Medications, and What Matters Most) [15,16,17], and community resources. Training on the geriatric syndromes covered: defining the syndrome, reviewing associated risk factors, why it was important to screen for the syndrome, and how to properly complete the related screening tool. Training was provided through online modules with pre- and post-tests to gauge additional learning needs, case-based applied learning sessions, sessions with community resource liaisons, and attendance at clinic-wide geriatric educational didactics. The online learning modules integrated content on common geriatric syndromes and the 4Ms (e.g., dementia and depression/Mentation; fall/Mobility; medication management/Medications; and What Matters Most).

CHWs were trained to screen for falls, depression, and cognitive impairment using validated tools [16,17,18,19] and/or questions developed and tested as part of the Indiana GETC program [9]. Following database training, CHWs recorded screening data in a Research Electronic Data Capture (REDCap) database along with education and resource referrals provided. REDCap is a Health Insurance Portability and Accountability Act (HIPAA)-compliant database created by Vanderbilt University [20]. The REDCap database had open-text fields for documenting narrative entries in response to participant-centered discussions directed by what the participant wished to focus on at the time of each encounter. The CHW took notes during each participant encounter and then entered a summary in the open-text REDCap fields of the topics and issues discussed with each participant immediately following each encounter. CHWs were required to pass all post-tests on core content with a score of at least 80% correct, to pass competency check offs on their ability to conduct patient-centered communication and motivational interviewing methods, and to accurately collect and document information in the REDCap database before beginning direct participant contact. 

### 2.2. Participants/Subjects 

The participants were 358 individuals with a mean age of 71.29 years (min = 65, max = 91). Further information regarding the participants can be found in Table 2. Participants were recruited from HealthNet (HN), a network of seven federally qualified health centers (FQHC). HN’s FQHC provides comprehensive health and health-related services to medically underserved persons in the Indianapolis area in the state of Indiana in the United States, with roughly 80% of participants living at or below the federal poverty line. All patients over the age of 65 and receiving care from an HN clinic were eligible for the study. 

### 2.3. Participant Recruitment and Enrollment

CHWs were deployed to cover the 7 FQHC and followed the recruitment and enrollment steps shown in Figure 1 (see below). 

Prior to March 2020, CHW encounters with participants were conducted face-to-face in clinics, at participant’s homes, or in the community. Due to COVID-19 restrictions, all CHW–participant contact moved to phone encounters during this March 2020 and August 2021 study period.

### 2.4. Data 

During the study period, 793 phone calls contacts were made. The average number of phone calls per patient was 4.89 (minimum =1, maximum = 30) excluding one outlier with 80 contacts due to an unusual set of urgent needs. Data from the open-text fields were extracted from all narrative entries and de-identified prior to performing analyses. 

### 2.5. Qualitative Analytic Methods

Qualitative analysis of all abstracted narrative entries was conducted following thematic analysis and crystallization immersion analytic methods based on the grounded theory [21,22,23]. These qualitative analytical methods were accomplished through completion of the following steps. MH and AS independently read through all the case notes and generated preliminary codes. All authors then came together to compare preliminary coding and reached consensus by resolving discrepancies between initial codes. A subset of the recoded data was discussed by all authors to ensure consistency in coding following the constant comparative method for developing coding schemes [24]. Emergent themes and subthemes were then generated using an iterative multi-step thematic content analysis [25]. After themes and subthemes were agreed upon, MH and AS selected representative entries for each subtheme. Authors DL and DB reviewed themes, subthemes, and representative quotes selected by MH and AS to assess the fit, relevance, workability, and modifiability of how well the concepts fit with the incidents before finalizing the themes and subthemes [26,27].

## 3. Results

### 3.1. Emergent Themes and Representative Quotes

Three overarching themes: social support systems, resource access, patient well-being/mood emerged from the data (see Table 3 below). 

#### 3.1.1. Social Support Structures

Social support structures were frequently mentioned in the case notes. Several subthemes emerged from the entries related to social support structures. Subthemes were receiving support from family, providing support to family, social support provided by non-family, and loneliness due to a lack of social support (Table 3). Participants reported being emotionally connected and supported by their children and grandchildren. Grandchildren were an important source of inspiration for living and gave purpose for those providing care for grandchildren. Reciprocally, friends became an important support structure for participants without adult children or with adult children who were unavailable. With our aging population, friends are often relied upon, especially when family members are unavailable. Losing friends became an ongoing reality for many older adults through the pandemic causing further isolation and lack of supports, and triggering fear of circumstances around one’s own death.

Over 80% of participants listed their children or grandchildren as their primary support system, and many participants either lived with their child/grandchild or had a child/grandchild living with them. In many cases, adult children or grandchildren acted as both the primary agent of social support and a central avenue for resource access. However, this support was often bidirectional. Participants in the program provided childcare, assisted with expenses, gave rides to appointments, and offered emotional support to children and grandchildren. This reciprocal altruism in parent–child and grandparent–grandchild relationships in aging families is well established [28]. However, tensions emerged in the data involving adult children and grandchildren providing support to program participants. Participants occasionally refrained from asking children for assistance due to fears of being a burden, and occasionally children were unable to meet the needs of participants due to geographic distance or scheduling conflicts. Additionally, a sort of transactional arrangements developed between some participants and their family members. This most frequently involved a dynamic where the participant relied on the family member for social support and the family member relied on the participant for resource access. This was more common when the participant lived in the home of a family member.

#### 3.1.2. Resource Access

Many participants required assistance with accessing resources. The most common resources that participants required assistance with were food, transportation, housing, medication, and medical supplies (Table 3). Participants were frequently unable to travel to the grocery store or unable to arrange for grocery delivery. Participants occasionally had access to food of low nutritional value but needed assistance to access healthy foods. Most content involving housing focused on assistance with moving (finding affordable housing, filling out required paperwork, etc.). The most common reasons for moving were being closer to family, stressful or dangerous living situations, and moving into a long-term care facility due to declining independence. Housing content, unrelated to moving, involved structural issues and difficulty paying rent or utilities. Calls involving medication almost uniformly involved difficulty getting prescriptions refilled, and most calls related to medical equipment dealt with mobility assistance (wheelchairs, walkers, scooters). 

In addition to being a standalone theme, access to transportation was also a key factor associated with the other resource-related themes. Lack of access to transportation was often the cause of food insecurity or difficulty refilling prescriptions. Transportation-related issues included friends or family being unavailable to assist with transportation, difficulty operating the technology required to access transportation (e.g., Lyft), and frustration with Medicaid-provided transportation services. 

#### 3.1.3. Patient Well-Being/Mood

Many entries documented patients’ well-being and/or mood. Negative patient moods were associated with effects of the Covid-19 pandemic, loneliness, and thoughts of mortality. Positive moods were associated with social interaction and communication with the CHW (Table 3). The COVID-19 pandemic was associated with fear, anxiety, sadness, and nervousness. Loneliness was associated with fear and sadness. Thoughts of mortality were associated with sadness. Social interaction was associated with happiness and contentment. Communication with the CHW was associated with happiness, relief, and gratefulness. 

For many participants, difficulty accessing these resources was tied directly to mental health and well-being concerns. Specifically, transportation-related issues were associated with feelings of depression and anxiety among participants, housing problems were associated with feelings of depression and anxiety, and difficulty refilling medications was associated with anxiety. Based on these findings, there is a clear relationship between resource access and the mental well-being of older adults. 

Limited access to resources contributed to both depression and anxiety in our study participants. For participants who did not have access to a vehicle, the loss of independence was associated with feelings of sadness and depression. Difficulty reaching the grocery store, pharmacy, and medical appointments was a source of anxiety in this group. Some participants who still drove experienced anxiety while operating their vehicles but continued to drive due to a lack of other transportation options. Uncertainty and insecurity regarding medication access was also a source of anxiety for participants in the program.

Many participants in the program required housing assistance, but this issue was particularly common among participants with a history of substance use disorder. Individuals in the program with a history of substance use disorder generally required more assistance from the CHW in finding stable housing. Moving (both before and after the move) was associated with acute anxiety among participants, and the desire to move but feeling unable to find alternative housing options was associated with depression.

### 3.2. The Complex Role of Family for Social Support and for Resource Access

For many participants, social support and resource access had the same source: children and grandchildren (Table 4). Many participants who reported reliable social support from children and/or grandchildren could also depend on these family members for assistance with resource access. Many children and grandchildren delivered groceries, picked up prescriptions, and gave rides to appointments in addition to providing social and emotional support. However, the converse of this was also true. Participants who reported a lack of social and/or emotional support from family also frequently had difficulty accessing resources. Additionally, the relationship between social and/or emotional support and resource access was sometimes transactional between participants and their children or grandchildren. This usually involved the patient providing resources (usually money, sometimes transportation) to the family member, who provided the patient with social and emotional support.

Issues related to resource access were typically rooted in diminished community mobility. Regardless of the cause of limited community mobility (inability to drive, lack of a personal vehicle, difficulty navigating public transportation), the most common strategy employed by participants was to utilize a proxy with a higher level of community mobility. The proxy, most frequently a child or grandchild, would perform tasks such as grocery shopping and refilling prescriptions on behalf of the participant to compensate for their difficulty with traveling. This person also generally provided transportation for medical appointments, court dates, and other events that required the participant’s attendance. Participants without a transportation proxy relied on ride sharing apps or public transportation but frequently had difficulty using these services.

### 3.3. Impact of COVID-19 on Older Adults

The COVID-19 pandemic had a cross-cutting impact identified in each of the three emergent thematic areas. The COVID-19 pandemic was a significant disruptor of both social support networks and resource access avenues, particularly among participants who relied heavily on family members for support and assistance. Participants reported an inability to see family or apprehension about seeing family due to COVID-19, fear and anxiety surrounding COVID-19, and difficulty accessing resources as a result of COVID-19. The effect of COVID-19 on resource access was twofold: COVID-19 increased the stress and demand on current resource access avenues, and it also disrupted the normal function of resource access avenues (Table 5). Participants frequently weighed the advantages of having visitors or leaving their home against the fear of potentially contracting COVID-19. Participants also frequently mentioned the precautions they were taking to avoid COVID-19.

COVID-19 and the health risks it posed to older adults were a source of fear and anxiety for many participants in the program. Infection among family members and risk of exposure made the transportation proxy model less practical and reliable for many participants, and many participants voiced an increased reluctance to leave the house. The telephone-based CHW service was able to meet the challenges posed by the pandemic by coordinating resources in a way that minimized in-person contact.

### 3.4. The Role of Community Health Workers in the Health and Well-Being of Vulnerable Older Adults

The roles that CHWs provide in filling gaps in patient needs left by standard primary care and family support systems were identified. CHW assistance was an important alternative to a transportation proxy for many participants in the program. CHWs were able to arrange grocery and medication delivery, assist with navigating public transportation, and arrange transportation for participants when necessary. This allowed participants with diminished community mobility to access resources without relying on a family member to meet their personal needs.

One of the more common roles of the CHW in the program was coordinating logistics related to moving. The moves frequently involved both the participant and children/grandchildren to some degree. The reasons for moving were varied but included moving to be closer to family, moving into a retirement community to ease the burden placed on family members, and moving out of a toxic living relationship with a family member. Housing instability was a prominent problem among participants with a history of substance use disorder (SUD). Complications of active substance use, past evictions, and past felonies made it difficult for older adults with SUD to find stable housing. CHWs assisted with paperwork, coordinated logistics with social workers and landlords, and helped participants find affordable housing that fit their needs. As a result of moving, participants reported feeling safer and less anxious. For those with caregivers, moving also eased some of the workload experienced by caregivers.

Adult children and grandchildren of participants were frequently able to provide both social support and resources to program participants. Reciprocally, participants without a supportive family structure often lacked both social support and assistance. This affected both the mental and physical health of this subgroup of participants, who were more likely to report feelings of loneliness and had greater difficulty obtaining healthy food and refilling medications. The CHW was able carry out some of the responsibilities typically shouldered by an adult child or grandchild

## 4. Discussion

Our data supported previous research [3,4,8] suggesting a relationship between social support systems and psychological distress as well as the relationship between low socioeconomic status and mental well-being among older adults from traditionally underserved populations. Considering the established relationships between social support, economic circumstances, and mental health (depicted in Figure 2 reproduced with permission) [29], a whole-patient approach is likely to be needed to address wellness and the health promotion of older adults from traditionally underserved populations.

The emergent themes in this study illustrate the densely intertwined relationships between older adults’ families, resource access avenues, and social support networks. The participants in this study expressed a wide variety of needs in terms of both type and severity exacerbated by the COVID-19 pandemic. In facing reduced access to the healthcare system, traditionally underserved populations typically experience relatively limited access to resources of all types. This was seen among G-care participants, who struggled to find transportation, housing, and nutrition assistance in addition to difficulties accessing the healthcare system. 

Our data also provide insights regarding the relationship between resource access and mental well-being. The experiences of the participants in the G-care program suggest that resource access, social support, and mental well-being are intertwined for older adults. For the older adults from traditionally low-income, underserved populations in this study, this relationship seemed particularly important to mental well-being. Overall, our qualitative analysis supports a whole-patient approach to mental health for older adults. 

Many of the concerns voiced by participants to CHWs fell within blind spots of conventional medical care for older adults. Assistance with moving, transportation (both medical and non-medical), and assistance with grocery shopping are not generally included in standard medical care, but these services are important in improving health-related outcomes in older adults. Appropriate living spaces are a key factor in maintaining independence and preventing falls, while transportation and access to healthy foods are important for maintaining proactive healthy behaviors. Considering the potential of these services in reducing morbidity and mortality among older adults, incorporating CHWs into standard health care delivery teams focused on vulnerable older adults would likely improve outcomes for both mental and physical health. 

One limitation of this study is that it was conducted within one health system in the Midwestern region of the United States, which potentially limits the generalizability of its findings. However, participants came from seven different FQHC across the health system, which provided the study with a diverse group of participants. Another limitation is that COVID restrictions limited all data to collection over the phone. While CHWs were still able to assist participants via the telephone, it restricted the amount of potential data gathered from participants. Additional data can be gathered in face-to-face visits where factors such as fall risks, food security, and living conditions can be visually assessed. An additional limitation was that the notes used for qualitative analysis reflect the CHWs’ interpretation of the older adults’ experience. Despite this limitation, this qualitative data repository provided a unique perspective based on the CHW’s scope of practice and role in the care continuum. The next step for this research is to correlate qualitative findings with quantitative measures collected on survey screeners (falls, depression, dementia, etc.). Analysis of these data is currently underway.

## 5. Conclusions

We found that CHWs are able to carry out responsibilities conventionally belonging to adult children and/or grandchildren of older adults, which allows for reliable resource access for participants with and without children while lessening caregiver burden. Evidence supports the view that services provided by CHWs are both clinical and cost effective [11,12,13,14]. This study supports the importance of incorporating CHWs in care management models for improving the health and well-being of vulnerable older adults.

## Figures and Tables

**Figure 1 ijerph-20-02766-f001:**
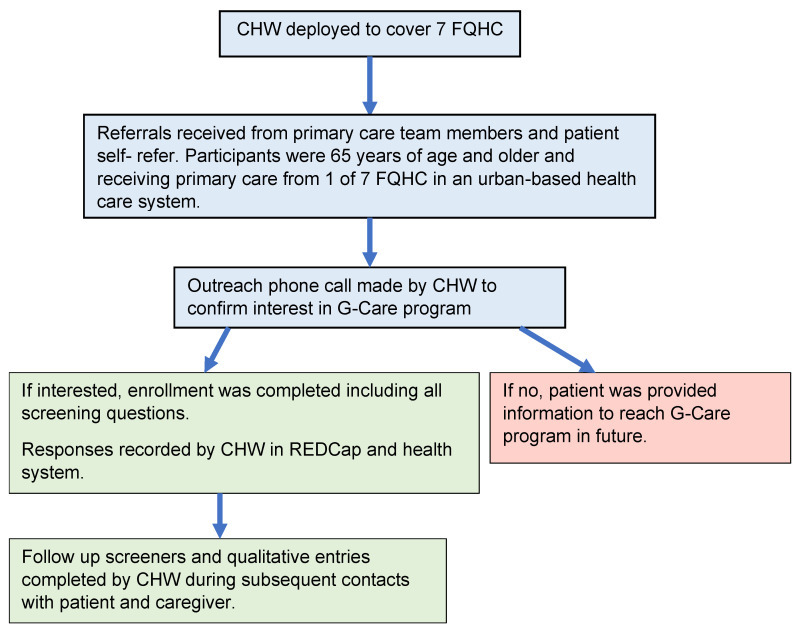
Recruitment and Enrollment Process.

**Figure 2 ijerph-20-02766-f002:**
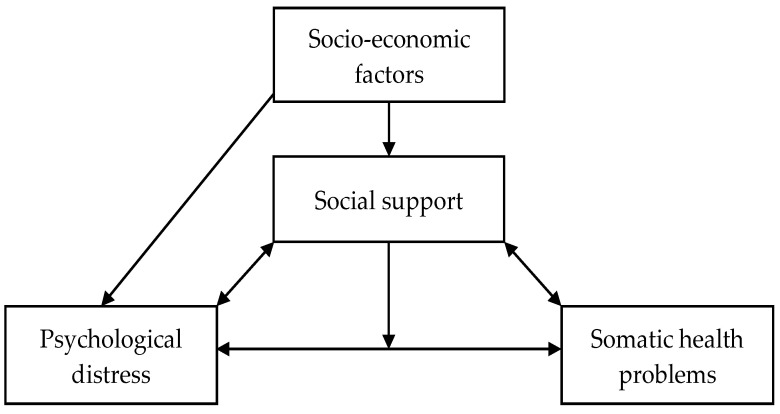
Conceptual model of the relationships between social support, psychological distress, somatic health problems and socio-economic factors [29].

**Table 1 ijerph-20-02766-t001:** Summary of CHW Training.

Community Health Workers (CHW) Trained by an Interdisciplinary Team Training included didactic, case-based learning, and online self-paced modules. Training covered: ○ Age-Friendly Framework (4Ms: Mentation; Mobility; Medications; What Matters Most) ○ Geriatric Syndromes and screening measures (e.g. falls, depression, cognition, risky medications, and opioids) ○ Community resources ○ Patient and caregiver educational information ○ Patient-centered communication/Motivational Interviewing Methods

**Table 2 ijerph-20-02766-t002:** Subject Demographics.

Total	358
Gender	
Male	29.7%
Female	70.3%
Ethnicity	
Hispanic	2.4%
Non-Hispanic	97.6%
Race	
White	49.5%
Black	46.7%
Other/Choose not to answer	3.8%

**Table 3 ijerph-20-02766-t003:** Emergent Themes from Qualitative Analysis with Representative Quotes.

I. Theme: Support Structures	Representative Quotes
Receiving Support from Family	“Participant (pt) loves her daughter in law and her grandchildren, and she can’t wait for them to come and visit.”
	“… pt does have some family support (son)… pt says she dealing with depression and learning to cope;… pt now has a new little granddaughter whom she feels gives her a reason”
	“pt has broken hip and depression; pt daughter Missy is caregiver and resides in the home”
Providing Support for Family	“pt doing well and is currently still the care giver for her grandchildren.”
	“pt is the caregiver of teen granddaughter while mother is away in army pt also has an adopted son age 11 from her brother she has been raising since he was a few days old.”
Non-family support systems	“pt having problems… pt has put son in jail (protective order) since last contact… pt has no family support just friends”
Loneliness	“pt lives alone with no family support or friends… pt states he was in fear of being alone or found dead in home after close friend passed;… pt has no one to call and check on him.”
II. Theme: Resource Access	Representative Quotes
Food	“pt at times is low on food with no assistance and always need someone to talk to”
Transportation	“Pt called geriatric coach regarding transportation to the grocery store… pt states she used a Lyft in the past but forgot how to operate the app”
	“pt has been a little depressed lately due to not being able to drive his vehicle”
	“pt needs transportation information and CICOA [local Area Agency on Aging office]; pt does drive but gets anxiety when in heavy traffic or traffic jams; pt states she can’t drive when weather changes”
Housing	“pt moved during the flood and has no family support in the city… pt states she needs assistance with housing and is having a hard time with bills”
	“pt is having housing issues and no longer feels safe; pt also has a history of substance abuse and wants help; pt ask that she be assisted with resources and senior housing”
	“pt wants to find a house asap; pt feeling depressed because time is winding down she stated she doesn’t feel safe staying another year and doesn’t want to moved back to Kentucky”
	“Pt contacted CHW for assistance with moving; pt is having anxiety and kind of scared to be on her own … pt asked that CHW get information for senior first floor living
Medication	“pt is recovering from substance abuse; pt needs assistance with medication refills and FSSA [Famiy and Social Services Administration]”
	“… the delivery of medication will help so much”
	“pt…is having trouble with risky medication; pt states doctor stopped prescription… upset and having anxiety.”
Medical Equipment	“pt currently lives with daughter on 3rd floor and pt is in need of wheelchair”
III. Theme: Patient Well-being/Mood	Representative Quotes
Positive	“Pt is happy everything is falling into place with the help from the G-care coach.”
Negative	“… during holidays pt gets down and depressed thinking about deceased son”
	“Pt is dealing with panic attacks and anxiety due to the fall… pt feel that anytime she goes outside she’s going to fall”

**Table 4 ijerph-20-02766-t004:** Complex Role of Family for Social Support and Resource Access.

Complex Role of Family	Representative Quote
Family/Transportation	“pt stated husband has drug problem and has been gone in her car for over 2 days; pt started crying but state she’s ok and has food; pt stated she just needs someone to talk to”
	“it’s hard for pt to receive transportation for her Dr. appointments… she doesn’t like to bother her children for rides”
	“Pt’s granddaughter is pregnant right now and taking her to her appointments is starting to be too much”
Family/Food/Transportation	“pt lives alone with no family support or friends… pt has an emotional support animal… pt needs assistance with food and transportation a lot”
Family/Housing	“pt was looking for housing assistance and needed help; pt lost wife and lives alone …he’s had previous drug problems and sometimes needs to speak with someone; pt has family support but doesn’t like bothering children”
	Pt… ” let her son borrow some money and he is not willing to pay her back before Thursday… she knows she needs the money to move into new apartment”.

**Table 5 ijerph-20-02766-t005:** Impact of COVID-19 on Support Structures, Resource Access, and Well-being.

Impact of COVID-19	Representative Quote
Family and Covid	“participant (pt) lives alone but has family support at this time… concerned about family visit and keeping grandchildren during COVID-19”
	“pt was upset… she has several family members in hospital at this time with COVID-19 pt… she feels down because there’s nothing she can do”
	“pt has concern because family member went for [COVID-19] testing and he need testing too but pt has no symptoms; pt just feeling anxiety due to the information the family member gave him”
Fear and Anxiety due to Covid	“Pt works at the high school so she felt a little better, but still scared of the COVID-19”
Medication and Covid	“pt needs assistance with medication pick up and information Prior to COVID-19 pt was going to Edna Martin for activities a few days a wk”
	“lately she is having a difficult time picking up prescriptions from pharmacy. G care coach informed patient that her pharmacy is delivering certain prescriptions to pts due to the COVID-19”
Work and Covid	“she doesn’t feel comfortable returning back to work due to her health conditions … she has been off for two months due to the COVID-19”
Transportation and Covid	“… called G care coach regarding transportation for Covid test”
Covid and Isolation	“pt says he is practicing safety and staying in due to COVID”

## Data Availability

Qualitative data are available from the corresponding author upon request.

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
