# Peer review of "The Role of Community Health Workers in the Health and Well-Being of Vulnerable Older Adults during the COVID Pandemic"

_ijerph, 2023, doi:10.3390/ijerph20042766_

Round 1
Reviewer 1 Report
Well-written paper with implications for gerontology and public health. Good sources and citations. Themes are adequately described and supported with the right amount of quotes from the data.
My only suggestion for improvement is to rework the abstract. Currently it seems to begin in the middle (with 2 incomplete sentences). I suggest starting with the sentences setting up the research problem, then moving to research design and results.
Author Response
|
Reviewers’ comment |
Description of revisions |
Location of revisions |
|
Reviewer # 1 comments:
|
|
|
|
My only suggestion for improvement is to rework the abstract. Currently it seems to begin in the middle (with 2 incomplete sentences). I suggest starting with the sentences setting up the research problem, then moving to research design and results. |
Abstract reworked using recommendations from reviewer #2.
|
See reviewer #2 |

Reviewer 2 Report
This is an article about the important role that CHWs played during the COVID-19 pandemic for older adults in Indianapolis. This is a very well-written article that clearly outlines the reason for the study, the methods, results, and discussion. The article needs only minor revisions for publication, mostly in the methods section, which I have outlined here:
1) I suggest moving the sentence on lines 17-18 in the abstract so it becomes the first sentence, which sets the stage for why this project was done.
2) Likewise, the sentence on lines 19-20 should be moved so that it is the concluding sentence in the abstract.
3) Line 132, change “refer to Figure 1” to “see below” and remove everything after that period.
4) Figures 1 and 2: Excellent graphics, thanks for including!
5) The authors will need to expand the Qualitative Analytic Methods section a bit:
a. How were the phone calls transcribed into these narrative entries? Were the calls audio recorded and transcribed later? By the same person? Or were entries completed during the phone call?
b. Line 149: Please describe further what thematic analysis, the crystallization immersion analytic method, and grounded theory are for the reader.
6) Lines 164-165: change your parenthesis to say “see Table 3, below).”
7) I appreciate the tables of themes and quotes, this is really well done, as is the results section.
Author Response
|
Reviewer # 2 comments:
|
|
|
|
I suggest moving the sentence on lines 17-18 in the abstract so it becomes the first sentence, which sets the stage for why this project was done. |
Sentence on lines 17-18 moved to first sentence
|
Line 11-12 |
|
Likewise, the sentence on lines 19-20 should be moved so that it is the concluding sentence in the abstract. |
Sentence on line 19-20 moved to concluding sentence |
Line 24-25 |
|
Line 132, change “refer to Figure 1” to “see below” and remove everything after that period. |
Changes made
|
Line 153 |
|
Figures 1 and 2: Excellent graphics, thanks for including! |
Thank you
|
|
|
How were the phone calls transcribed into these narrative entries? Were the calls audio recorded and transcribed later? By the same person? Or were entries completed during the phone call? |
Entries were made by CHW after each participant encounter in the open-text fields in the REDCap database. A clarifying sentences was added to explain this. |
Line 129-131. |
|
Line 149: Please describe further what thematic analysis, the crystallization immersion analytic method, and grounded theory are for the reader. |
A sentence was added indicating that the paragraph on qualitative methods includes the series of steps used to accomplish the crystallization immersion analytic method. |
Lines 173-74. |
|
Lines 164-165: change your parenthesis to say “see Table 3, below).” |
Changes made |
Line 189 |
|
I appreciate the tables of themes and quotes, this is really well done, as is the results section. |
Thank you |
|

Reviewer 3 Report
Dear authors,
I would like to thank you for the opportunity to review your manuscript.
The manuscript is well written, the results are well presented and discussed.
I have few minor suggestions:
1. Your abstract begins with the research goals (and with the word "to"). I would add some background before describing the research goals (preferably a background that leads to the research questions).
2. In the "Materials and Methods" chapter, lines 98-99: you describe common geriatric syndromes included in the training (the 4Ms, age-friendly frameworks, and community resources). I think it would be better and clearer to provide with more detailed description of these syndromes.
Author Response
|
Reviewer # 3 comments:
|
|
|
|
Your abstract begins with the research goals (and with the word "to"). I would add some background before describing the research goals (preferably a background that leads to the research questions). |
Abstract reworked using recommendations from reviewer #2. Sentence structure changed.
|
Lines 11-14; lines 24-25. |
|
In the "Materials and Methods" chapter, lines 98-99: you describe common geriatric syndromes included in the training (the 4Ms, age-friendly frameworks, and community resources). I think it would be better and clearer to provide with more detailed description of these syndromes. |
Clarification and a more detailed description of what training on the geriatric syndromes covered was added. |
Line 110-115 |
|
|
|
|
|
Acknowledgments |
Individuals who helped implement the G-Care program were added |
Line 423-24. |
